# Therapy Resistant Gastroenteropancreatic Neuroendocrine Tumors

**DOI:** 10.3390/cancers14194769

**Published:** 2022-09-29

**Authors:** Kristen McClellan, Emerson Y. Chen, Adel Kardosh, Charles D. Lopez, Jaydira Del Rivero, Nadine Mallak, Flavio G. Rocha, Yilun Koethe, Rodney Pommier, Erik Mittra, Guillaume J. Pegna

**Affiliations:** 1School of Medicine, Oregon Health & Science University, Portland, OR 97239, USA; 2Division of Hematology Oncology, Knight Cancer Institute, Oregon Health & Science University, Portland, OR 97239, USA; 3Developmental Therapeutics Branch, Center for Cancer Research, National Cancer Institute, National Institutes of Health, Bethesda, MD 20892, USA; 4Division of Molecular Imaging and Therapy, Oregon Health & Science University, Portland, OR 97239, USA; 5Division of Surgical Oncology, Department of Surgery, Oregon Health & Science University, Portland, OR 97239, USA; 6Dotter Department of Interventional Radiology, Oregon Health & Science University, Portland, OR 97239, USA

**Keywords:** gastroenteropancreatic neuroendocrine tumors, resistance, neuroendocrine tumors, peptide receptor radionuclide therapy, somatostatin receptor, targeted therapies, chemotherapy

## Abstract

**Simple Summary:**

The medical care for patients with neuroendocrine tumors arising from the intestines, pancreas, or stomach is frequently limited by the development resistance to common treatment approaches including targeted therapies and chemotherapy. The purpose of this review is to summarize the current treatments for these tumors, the possible cellular changes involved with the development of treatment resistance, and possible ways to overcome or approach this challenge. The goal is to provide an up to date summary of current and upcoming clinical findings regarding therapy-resistant neuroendocrine tumors.

**Abstract:**

Gastroenteropancreatic neuroendocrine tumors (GEP-NETs) are a heterogenous group of malignancies originating from neuroendocrine cells of the gastrointestinal tract, the incidence of which has been increasing for several decades. While there has been significant progress in the development of therapeutic options for patients with advanced or metastatic disease, these remain limited both in quantity and durability of benefit. This review examines the latest research elucidating the mechanisms of both up-front resistance and the eventual development of resistance to the primary systemic therapeutic options including somatostatin analogues, peptide receptor radionuclide therapy with lutetium Lu 177 dotatate, everolimus, sunitinib, and temozolomide-based chemotherapy. Further, potential strategies for overcoming these mechanisms of resistance are reviewed in addition to a comprehensive review of ongoing and planned clinical trials addressing this important challenge.

## 1. Introduction

This Special Issue of *Cancers* is focused on identifying and targeting molecular pathways of resistance, reporting original clinical research and reviewing the clinical management of gastroenteropancreatic neuroendocrine tumors ‘beyond the first line’. Neuroendocrine neoplasms (NENs) comprise a diverse group of malignancies that arise from neuroendocrine cells originating from any organ in the human body. As such, NENs are further described by their histologic morphology and primary site of origin. The majority of NENs are well-differentiated and classified as neuroendocrine tumors (NETs), while 10–20% are poorly differentiated and referred to as neuroendocrine carcinomas (NECs) [1]. While the majority of NETs are of embryologic midgut origin, the most common primary tumor sites include those originating from the gastroenteropancreatic organs (GEP-NETs) [2]. Pathologically, NENs demonstrate immunohistochemical markers consistent with neuroendocrine differentiation including chromogranin A, synaptophysin, and/or neuron-specific enolase [3]. While great strides have been made in the diagnosis, management, and treatment of GEP-NETs, treatment resistance remains a critical challenge, particularly considering the limited therapeutic options available for these tumors. Treatment resistance in GEP-NETs may be observed both as primary resistance, or inherent resistance to a treatment manifested as the lack of any response, or secondary resistance, meaning the loss of efficacy in spite of a prior response [4].

## 2. GEP-NET Epidemiology, Pathology, Grading, and Staging

While frequently considered a rare tumor, the incidence of NETs has rapidly increased over the past few decades, especially in primary sites including the lung, small intestine, stomach, and rectum [2,5]. From 1973 to 2012, cases rose 6.4-fold in the United States, to an estimated annual incidence in 2012 of 6.98 cases per 100,000 people [2]. This rise in incidence has largely been seen in the diagnosis of locoregional rather than metastatic disease. This observation suggests that that this increase in incidence is at least in part due to improved detection of these diseases via increased imaging and endoscopic procedure frequency and improved sensitivity of imaging technologies [2,6,7,8]. 

While NETs can originate from any organ, 60.9% of primary sites involve the gastrointestinal tract followed by 27.4% from bronchopulmonary sites [3,9]. The most common of these, GEP-NETs, include those originating from the small intestine (30.8%), rectum (26.3%), colon (17.6%), pancreas (12.1%), stomach (8.9%), and appendix (5.7%) [9]. At times, the primary site of origin cannot be identified in spite of a comprehensive workup, in which case these are referred to as NETs of unknown origin. 

In 2019, the World Health Organization (WHO) implemented a histologic classification system for GEP-NENs. Per this classification scheme, all poorly differentiated GEP-NENs are neuroendocrine carcinomas (NECs) and classified as high grade [10]. GEP-NETs are all well-differentiated but divided into low, intermediate, and high grade based upon the tumor’s mitotic rate as described by the mitotic or Ki-67 indices. Tumors are low-grade (G1) if they have a mitotic rate < 2 mitoses/2 mm^2^ or Ki67 index <3%, intermediate-grade (G2) if the mitotic rate is 2–20 mitoses/2 mm^2^ or Ki-67 index is 3–20%, and high-grade(G3) when the mitotic rate is >20 mitoses/2 mm^2^ or Ki67 index is >20%, with relatively worse prognoses observed at Ki67 index >55% (Figure 1) [10,11]. While the WHO histologic classification and grading system for GEP-NENs has been widely adopted, studies have suggested intra-grade heterogeneity with regard to prognosis. This appears to be particularly notable within grade 2 disease, in which those patients with Ki-67 index 3–9% may have improved survival compared to those with Ki-67 index 10–20% [12].

Imaging of these tumors is now commonly performed with functional positron emission tomography (PET) with either/both radiolabled somatostatin receptor (SSTR) anologues or fluorodeoxyglucose (FDG). Commonly utilized radiolabled somatostatin anologues include the somatostatin anologue octreotate, covalently bonded by the bifunctional chelator tetraxetan (DOTA) to the radioisotopes Gallium-68 (^68^Ga oxodotreotide or DOTATATE) or Copper-64 (^64^Cu oxodotreotide or DOTATATE), or the somatostatin anologue Tyr3-octreotide, covalently bound via the DOTA chelator to radioisotopes Gallium-68 (^68^Ga edotreotide or DOTATOC) or Copper-64 (^64^Cu edotreotide or DOTATOC) [13]. 

On functional imaging, low and intermediate grade tumors most commonly demonstrate elevated SSTR-directed and poor FDG PET avidity. Taken together, these findings are thought to be reflective of the more common expression of SSTR and lower metabolic activity observed in these tumors. Conversely, high grade NETs and NECs frequently demonstrate elevated FDG PET and poor SSTR-directed PET avidity, thought to be reflective of the increased metabolic activity and decreased SSTR expression in these tumors (Figure 1) [1,14].

Staging of NETs is performed as per the American Joint Committee on Cancer tumor, node, and metastasis (TNM) staging system and differs based on the primary disease site [1,14]. The most common sites of metastasis are lymph nodes, liver, mesentery, peritoneum, lung, and bone, with rare involvement of the brain [8].

### 2.1. GEP-NET Prognosis and Clinical Manifestations

The prognosis for GEP-NETs is most directly determined by tumor grade, stage, and primary disease site. Small intestine NETs (siNETs) are classically considered to have a better prognosis than pancreatic NETS (panNETs) or colorectal NETs, although when controlled for tumor grade, this difference is less clear [1]. Higher grade and/or stage tumors are consistently associated with worse prognosis [14]. 

GEP-NETs may present with a broad diversity of clinical manifestations and are frequently discovered incidentally. Factors contributing to clinical manifestations include mass effect, tumor functionalilty, and the primary site, size, and locations of metastastatic disease [8]. Tumor functionality refers to the rather unique ability of NETs to secrete hormones, a characteristic reflective of their cellular origin. This ability may result in symptoms or clinical syndromes related to the specific hormone secreted. Within siNETs, approximately 19% secrete serotonin or other related vasoactive substances, which may result in the development of carcinoid syndrome, particularly in the setting of bulky liver metastases [3,15]. Carcinoid syndrome presents with wheezing, skin flushing, abdominal cramping, diarrhea and eventual right-sided cardiac valvular fibrosis [8]. Only 10–30% of panNETs are functional, secreting insulin (insulinoma), glucagon (glucagonoma), gastrin (gastrinoma), vasoactive intestinal peptide (VIPoma), or somatostatin (somatostatinoma) [8,16]. Recurrent hypoglycemia is associated with insulinomas; epigastric pain, diarrhea, and ulcer disease with gastrinomas; hyperglycemia, weight loss, and necrolytic migratory erythema with glucagonomas; secretory diarrhea, metabolic acidosis, and hypokalemia with VIPomas; and diabetes mellitus, cholelithiasis and steatorrhea with somatostatinomas [1,3,8,16]. Utilized within the appropriate context, the aforementioned hormones as well as urinary 5-Hydroxyindoleacetic Acid (5-HIAA) for patients with carcinoid syndrome can be utilized for the diagnoses of these tumors and the associated hormonal syndromes. Symptoms from non-functioning GEP-NETs are often a result of mass effect or nonspecific and may present as abdominal pain, nausea, vomiting, jaundice, early satiety, weight loss, and/or gastrointestinal bleeding [8,16].

### 2.2. Genetics of GEP-NETs

The majority of GEP-NETs occur in the setting of sporadic mutations and exhibit low mutation rates compared to other solid tumor malignancies [8,17]. The genetic mutations identified most commonly involve regulation of the mammalian target of rapamycin (mTOR) pathway, epigenetic changes (including chromatin modification, telomere length regulation, and others), cell cycle, and DNA repair pathways [17,18]. The most common somatic mutations in panNETS include genes involved in the epigenetic machinery including multiple endocrine neoplasia (MEN) 1, followed by Death-associated protein 6 (DAXX) and ATP-dependent helicase (ATRX) [16,19]. While germline MEN1 mutations are pathognomonic to the hereditary syndrome, multiple endocrine neoplasia type 1, most somatic MEN1 mutations in GEP-NETs are sporadic, and are thought to play an important role in tumorigenesis [17]. ATRX and DAXX mutations are frequently observed, particularly in panNETs, and affect the epigenetic alternative lengthening of telomeres (ALT) pathway and chromosome instability [17,18,20,21]. Within siNETs, 20% have cyclin-dependent kinase inhibitor 1B (CDKN1B) mutations that result in cell cycle dysregulation [22,23].

Only 5 to 10% of GEP-NETs are associated with hereditary syndromes and of those, most involve the pancreas [8]. Hereditary syndromes associated with panNETs are often inherited in an autosomal dominant pattern and include multiple endocrine neoplasia type 1 (MEN1), von Hippel-Lindau syndrome (VHL), tuberous sclerosis 1 and 2 (TSC1 and TSC2), and neurofibromatosis type 1 (NF1) [8].

### 2.3. Current Treatment of GEP-NETs

Treatment of GEP-NETs generally depends upon tumor primary site, grade, and staging. For local or locoregional disease, surgery remains the cornerstone of management, allowing for the greatest likelihood of optimal outcomes in non-metastatic disease [24]. Surgical interventions include enucleation, resection, lymphadenectomy, and/or cytoreductive operations [14]. Other locally targeting modalities include radiofrequency ablation or liver-directed therapies such as chemoembolization, arterial embolization, radioembolization, or ablative therapy [3,14].

For advanced, inoperable or metastatic GEP-NETs that express the somatostatin receptor (SSR+), the first line treatment consists of the somatostatin analogues, octreotide acetate or lanreotide [25,26]. Second-line systemic therapies for both siNETs and panNETs include both peptide radionuclide receptor therapy (PRRT) with Lutetium-177 DOTATATE (^177^Lu-DOTATATE) for SSTR+ disease and everolimus [27,28]. Additional therapies have been utilized specifically for panNETs in the second and later line setting include sunitinib and temozolamide-based chemotherapy [29,30]. Immune therapies, particularly the programmed death protein 1/programmed death-ligand 1 (PD-1/PD-L1) and cytotoxic T-lymphocyte associated protein 4 (CTLA-4) immune checkpoint inhibitors have demonstrated limited activity which has been primarily demonstrated in high-grade disease (grade 3 NET and NEC) [31]. While treatment options for GEP-NETs remain limited, multiple clinical trials are ongoing investigating novel therapies and therapeutic combinations and comparing current modalities in order to optimize the management of the complex diseases [29,30]. While this research remains ongoing, the development of resistance to existing treatment options for GEP-NETs remains a significant challenge in the management of these patients. 

## 3. Treatment Resistance in GEP-NETs

### 3.1. Somatostatin Analogues: Octreotide Acetate and Lanreotide

The somatostatin analogues (SSAs), octreotide and lanreotide, exert their mechanism of action through binding SSTRs (G protein-coupled receptors) with strong affinity for SSTR2 and moderate affinity for SSTR5 [32,33]. Consistent with this mechanism of action, positive SSTR status by functional imaging or histology is required although not predictive of anti-proliferative and anti-secretory response to SSAs [14,34]. Octreotide acetate and lanreotide have demonstrated benefit in the treatment of NETs through both an anti-proliferative effect (as evidenced by decreased tumor growth) and an anti-secretory effect, resulting in improvement of symptoms of hormone hypersecretion in patients with functional tumors [25,26,30]. The anti-proliferative effect of SSAs is understood to be mediated through SSTR activation of protein tyrosine phosphatases (PTPs) and cGMP-dependent protein kinase, leading to the regulation of multiple downstream signaling pathways, resulting in decreased cell growth, proliferation, and migration/invasion [35,36,37,38,39]. In contrast, the anti-secretory effect is believed to be mediated through SSTR activated G protein-mediated inhibition of voltage-dependent calcium channels and the stimulation of potassium channels, thus inhibiting hormone secretion [40,41]. Both SSAs may be administered as long-acting, monthly intramuscular (octreotide acetate) or subcutaneous (lanreotide) injections [25,30].

The superiority of octreotide acetate over placebo in patients with metastatic, well-differentiated GEP-NETs was demonstrated in the PROMID trial (NCT00171873), published in 2009. This prospective, randomized, double-blinded study demonstrated significantly improved median time to tumor progression with octreotide compared to placebo (14.3 vs. 6 months, respectively, *p* < 0.001) [26]. Similarly, the CLARINET trial (NCT00353496) published in 2014, was a randomized, double-blind study in patients with metastatic, grade 1 or 2, non-functioning, enteropancreatic NETs. This trial demonstrated improved progression-free survival (PFS) at 18 and 24 months in patients receiving monthly lanreotide compared to those receiving placebo (median not reached vs. median of 18 months, respectively, *p* < 0.001) [25]. The low objective response rates to octreotide acetate and lanreotide (2.4, 2%, respectively), supports the notion that both agents act through cytostatic rather than cytocidal mechanisms. The anti-secretory effects of both SSAs have similarly been supported in prospective clinical trials utilizing these agents for the treatment of carcinoid syndrome [42,43]. Predictors of decreased time to progression to SSAs include primary pancreatic site of disease, presence of liver metastases and higher grade tumors, while initiation of treatment in the presence of stable disease, male sex, and carcinoid heart disease are predictive of a longer response [44]. The US Food and Drug Administration (FDA) approved octreotide for symptomatic treatment of carcinoid syndrome in 1998 and lanreotide in 2014 for patients with unresectable, grade 1 or 2, metastatic GEP-NETs.

Consistent with the dual anti-proliferative and anti-secretory benefits of SSAs for GEP-NETs, resistance to these agents can be manifested through both tumor progression and recurrence of hormonal symptoms. Within the context of carcinoid syndrome, resistance or insufficient anti-secretory response to SSAs in clinical trials has been defined as experiencing an average of four or more bowel movements per day, in spite of receipt of stable-dose SSAs [45].

Proposed mechanisms for the development of resistance to octreotide or lanreotide include epigenetic changes within tumor cells including histone methylation and/or acetylation changes that impact SSTR expression (Figure 2) [46,47,48,49]. More recently, further epigenetic regulation of SSTR expression in PanNETs has been proposed to occur through the expression of the antisense transcript, SSTR5-AS1. Expression of this antisense transcript appears to be influenced by the methylation of selected CpG islands, and resulted in downregulated SSTR5 expression and decreased SSA response [50]. The importance of epigenetic changes to the development of SSA resistance has been supported by studies demonstrating the effects of tumor exposure to both histone deacetylase inhibitors (HDACi) and DNA methyltransferase inhibitors (DNMTi) resulting in increased SSTR2 expression in NET cells both in vitro and in small animal imaging models [49,51]. The DNMTi, decitabine, and HDACi, tacedinaline, also demonstrated increased SSTR2 expression in the NET cell lines, BON1, H727, and QGP1 [52]. Valproic acid, a HDACi, was found to upregulate SSTR2 expression and inhibit GEP-NET cell growth through regulating MYC signaling, TGF-β1, FOXO3, and p53 [53,54].

Overcoming this epigenetically derived resistance could potentially be achieved by combining SSAs with a HDACi and/or DNMTi to potentially augment the uptake and anti-tumor activity of these agents [47,48,49,54]. However, the toxicities of existing HDACi’s and DNMTi’s would provide significant challenges to implementing this approach. Agents from both classes are known to be capable of causing significant myelosuppression and bothersome gastrointestinal symptoms while HDACi’s have been associated with cardiac toxicities and ventricular arrhythmias [55,56].

Beyond epigenetic alterations, there is limited research within NETs regarding alternative resistance mechanisms to SSAs. Mechanisms have been studied in other cancer types that express SSTR [57]. In the neuroblastoma and glioma hybrid cell line, NG108-15, for example, cells exposed to continuous somatostatin become desensitized to the drug through internalization of the SSTR [58]. Both the rate and magnitude of desensitization were dependent on the concentration of somatostatin administered [58]. In these cells, the administration of an inhibitor of receptor sequestration led to reduced desensitization to somatostatin [58]. When pituitary tumor cells that overexpressed the SSTR2A somatostatin receptor were exposed to somatostatin or an SSA, desensitization mechanisms included receptor sequestration as well as receptor phosphorylation (Figure 2) [59]. Increasing the dose of SSAs or intermittent drug holidays may briefly reverse desensitization in cell models, however this approach has not demonstrated efficacy when attempted in NET patients resistant to SSA treatment [57]. Further, considering that resistance to SSAs typically develops over weeks or months, receptor internalization may not represent the predominant mechanism of resistance in these tumors. Rather, this time course may be consistent with the growth of tumor cells clones that do not express the targeted SSTR subtype [60]. 

Targeting alternative SSTRs present on NETs has been proposed as a potential means of overcoming resistance to existing SSAs. While octreotide acetate and lanreotide have a high affinity for SSTR2 and moderate for SSTR5, pasireotide, is a somatostatin analogue that exhibits a high affinity for binding SSTR1-3, and 5 [32,33]. Neuroendocrine tumors can markedly differ in their expression of somatostatin receptor subtypes, thus suggesting a possible mechanism allowing for pasireotide to overcome SSA resistance [33,60,61]. The efficacy of pasireotide was evaluated in a phase III study by Wolin et al. in patients with metastatic neuroendocrine tumors with carcinoid syndrome refractory to prior SSA. This study found that when compared to octreotide acetate 40 mg, pasireotide 60 mg (both given every 28 days), exhibited similar symptom control rates (20.9% vs. 26.7%, *p* = 0.53), a trend towards higher tumor control rates (62.7% vs. 42.6%, *p* = 0.09), and a longer progression-free survival (11.8 months vs. 6.8 months, *p* = 0.045) [32]. While these findings suggested potential benefit of pasireotide in this patient population, pasireotide has additionally been found to cause significantly more frequent hyperglycemia, nausea, and bradycardia than octreotide acetate, and has not been approved for use in NETs [32,62].

Alternative strategies to overcome the lack of response to the anti-secretory effects of SSAs in functional NETs include the addition of the tryptophan hydroxylase inhibitor telotristat ethyl in carcinoid syndrome as well as the second-line use of PRRT with ^177^Lu-DOTATATE. Telotristat ethyl was approved in 2017 in combination with SSA therapy for the treatment of adults with carcinoid syndrome diarrhea inadequately controlled by SSA therapy alone [63]. Telotristat ethyl is understood to have synergistic anti-secretory effects when combined with SSAs by inhibiting tryptophan hydroxylase (TPH), the rate-limiting enzyme in serotonin synthesis [45]. The second-line use of PRRT has similarly been shown to improve hormone-related symptoms in metastatic functional NETs refractory to treatment with SSAs. Of those considered resistant to octreotide therapy (as well as multi-kinase inhibitors and chemotherapy), those who received ^177^Lu-DOTATATE experienced a 54% complete and 35% partial response in clinical symptoms, and 17% complete and 28% partial response in chromogranin A biomarker levels [64]. Although the mechanism of maintained efficacy of a SSTR-directed treatment (PRRT) in patients resistant to somatostatin analogue therapy is unknown, it suggests that therapies targeting the SSTR remain viable targets for future research seeking to overcome resistance to octreotide and lanreotide.

### 3.2. Peptide Receptor Radionuclide Therapy (PRRT): ^177^Lu-DOTATATE

PRRT is a general therapeutic modality in which a radionuclide is linked to a peptide capable of targeting a receptor to deliver a localized radiopharmaceutical payload. ^177^Lu-DOTATATE is the first FDA-approved PRRT and utilizes a somatostatin analogue (DOTATATE) covalently bound to the beta-minus emitting radioisotope ^177^Lu in order to provide targeted radiation directly to NET cells overexpressing SSTRs (primarily SSTR2) [27,65]. This PRRT is administered intravenously generally every 8 weeks for a total of 4 cycles. Similarly to SSAs, SSTR positivity is mandatory for response to PRRT utilizing ^177^Lu-DOTATATE [14]. The efficacy of this therapy was confirmed in the NETTER-1 trial (NCT01578239) comparing ^177^Lu-DOTATATE, combined with the standard dose of 30 mg of octreotide versus 60 mg octreotide alone in patients with metastatic midgut NETs and radiographic progression on first-line SSA therapy [27]. NETTER-1 demonstrated that treatment with ^177^Lu-DOTATATE and octreotide resulted in a progression free survival (PFS) rate of 65.2% vs. 10.8% in the high-dose octreotide group. The objective response rate was significantly higher in the ^177^Lu-DOTATATE group at 18% vs. 3% in the octreotide group (*p* < 0.001) [27]. In 2021, Strosberg et al. provided updated survival results from the NETTER-1 trial, which showed no statistically significant difference in median overall survival in the ^177^Lu-DOTATATE group and octreotide group vs. the control group (48.0 months vs. 36.3 months, respectively, *p* = 0.30), a result that was likely impacted by the high rate of crossover (36%) in the investigational arm of the study [66]. ^177^Lu-DOTATATE was approved for use in advanced GEP-NETs by the FDA in 2018 [65]. Subsequent studies have identified multiple factors associated with unfavorable prognosis or response to PRRT. These include elevated inflammatory markers (including elevated C-reactive protein, neutrophil-to-lymphocyte, and platelet-to-lymphocyte ratio), the presence of ascites, marked liver metastases burden, ≥5 bone metastases, and FDG PET disease avidity [67,68,69,70,71]. In contrast, higher ^68^Ga-DOTATATE PET avidity and primary pancreatic disease site have been associated with improved response to PRRT [72,73].

While frequently used in the second-line setting following the development of resistance to SSA, resistance to PRRT may also occur. Waldeck et al. proposed that rather than target (SSR) down-regulation or alteration, PRRT failure appears to be related to enhanced DNA damage repair pathways resulting in rapid repair of DNA damage induced by the radionuclide emissions (Figure 2) [74]. In their study, Xenograft SSR2 cell lines (AR42J, H1299-7, and H69) were repeatedly exposed to ^177^Lu-DOTATATE until they developed resistance and then evaluated for double strand breaks by utilizing gamma H2Ax staining. The initial DNA damage from radiation was not different in the resistant cell lines in immunohistochemical staining, suggesting upregulated DNA repair in these cells [74].

An additional potential mechanism for the development of both primary and secondary resistance to PRRT is the activation of hypoxia signal pathways through true tissue hypoxia or by genetic influences that promote the activation of similar hypoxia-related pathways [75]. Demonstrated in mouse pheochromocytoma cells, the expression of hypoxia-inducible factor 2 alpha, or HIF-2α, was associated with increased resistance to ^177^Lu-DOTATATE (Figure 2) [75]. HIF-2α expression has been associated with the development of a radioresistant phenotype in other cancers including renal cell carcinoma, the mechanism of which is hypothesized to occur through interaction with the tumor suppressor, p53 [75]. While its use in combination with radiotherapies has not yet been studied, a HIF-2α inhibitor, belzutifan, has been demonstrated to be clinically effective in a phase 2 study of patients with von Hippel-Lindau syndrome and renal cell carcinoma (RCC). This study demonstrated that the use of belzutifan resulted in a 49% objective response for their RCC as well as a 77% response in those patients with synchronous with panNETs [76]. Future therapies combining a HIF-2α inhibitor such as belzutifan, could offer a potential treatment augmentation strategy to address hypoxia and HIF-2α mediated resistance to PRRT.

Another potential mechanism to improve the efficacy of PRRT in GEP-NETs and potentially overcome observed resistance would be the use of higher energy alpha-emitting radionuclides in place of currently utilized beta-emitting PRRT. Alpha-emitting PRRT would have the potential to cause significantly higher (estimated 20 fold) amounts of double strand DNA breaks due to the more localized, higher linear energy delivery provided by alpha particles to the targeted tissues [77]. Double strand DNA breaks effectively and rapidly induce cell apoptosis, which may allow for the prevention and circumventing of hypoxia signaling pathways and upregulated DNA repair mechanisms that could otherwise to lead to radiation resistance as is seen with ^177^Lu-DOTATATE [77]. A retrospective study of 7 patients with neuroendocrine tumors resistant to the beta-emitting radiopharmaceuticals, Yttrium-90 (^90^Y)/^177^Lu-DOTATOC, were treated with the alpha-emitting peptide receptor therapy, Bismuth-213 (^213^Bi)-DOTATOC. Of the 7 patients, objective responses were complete in 1, partial in 2, and stable in 3 patients as determined by RECIST (Response Evaluation Criteria in Solid Tumors) imaging criteria [78]. These results suggest a promising approach for the treatment of GEP-NETs resistant to beta-emitting PRRT. Current challenges to this potential strategy include cost, limited supplies for production, a short half-life, and hematological and renal toxicity [77,78]. Clinical trials are currently ongoing evaluating the safety and efficacy of the alpha emitting radiopharmaceuticals, Lead-212 (^212^Pb)-DOTAMTATE (octreotate covalently bound to the chelating agent DOTAM) and Actinium-225 (^225^Ac)-DOTATATE, in patients with metastatic and unresectable NETS, including in those patients who have progressed after initial PRRT treatment with ^177^Lu-DOTATATE (NCT05153772, NCT03466216, NCT05477576) (Table 1).

While the use of alpha emitting PRRT represents a particularly exciting area of clinical research to improve upon the clinical activity observed with ^177^Lu-DOTATATE, other PRRT modalities have also been evaluated including the use of Copper-64 sarcophagine octreotate (64Cu SARTATE), octreotate linked with a more stable radionuclide isotope ^64^Cu. In a prospective trial of 10 patients, the imaging of ^64^Cu-SARTATE exhibited tumor uptake with a statistically significant higher max lesion SUV_max_ at 1 h (41.9 vs. 31.65, respectively, *p* = 0.01) and 4 h (50.5 vs. 31.65, *p* = 0.004) when compared to 68-Ga-DOTATATE. ^64^Cu-SARTATE also demonstrated a safe profile with no grade 3 or 4 adverse events [79]. This more stable radionuclide isotope could serve as a promising PRRT therapy by prolonging activity at the site of pathology in patients with NETs, although no current clinical trials are evaluating this approach [79].

Improving NET radiation sensitivity through the use of adjunct therapies has been the goal of multiple clinical studies to date. These include studies demonstrating that capecitabine, capecitabine with temozolomide, everolimus, and 5-fluorouracil, can be safely administered with PRRT, although none have been powered to demonstrate superiority in therapeutic effect to PRRT alone [80,81,82,83,84,85,86]. More recent interest has developed in targeting the epigenetics of NETs in order to enhance radiotherapy sensitivity in these tumors. Epigenetic changes resulting in radiotherapy resistance are thought to occur either through the epigenetic changes already involved in NET carcinogenesis or the acquired changes from the effects of radiation itself as has been demonstrated to occur in other malignancies treated with radiation [87,88,89]. Within the context of mid-gut NETs, Pollard et al. demonstrated that patients undergoing PET ^68^Ga^-^DOTATOC imaging before and after treatment with the HDACi vorinostat demonstrated significantly increased DOTATOC avidity at metastatic hepatic sites, suggesting that vorinostat pretreatment may result in increased SSTR expression [90]. While further mechanistic and confirmatory studies are needed, these findings suggest that epigenetic modifiers could potentially be used to increase SSTR expression, opening a future avenue for addressing epigenetically derived PRRT resistance. 

Other radiosensitizers being studied for potential synergistic use along with PRRT include mTOR inhibitors and poly (ADP-ribose) polymerase (PARP) inhibitors. A study by Exner et al. found pre-treatment of five different neuroendocrine neoplasm cell lines (BON, QGP-1, LCC-18, H727, UMC-11) with an mTOR inhibitor, temsirolimus, provided improved in vitro sensitivity to irradiation [91]. Combined mTOR inhibitor and irradiation decreased cell numbers and survival in all cell lines when compared to temsirolimus and irradiation alone [91]. Similarly, potential anti-tumor synergy was demonstrated in a study by Cullinane et al. in which PRRT was combined with the PARP1 inhibitor, talazoparib. This combination was tested in vitro against SSR2 expressing AR42J cell lines compared to PRRT alone and was found to result in a significant increase in DNA double strand breaks and improved overall anti-tumor efficacy [92]. Similar results using the PARP1,5 inhibitor dihydroxyisoquinoline demonstrated that combining PRRT with this agent improved cytotoxicity in both gastroenteropancreatic (BON-1) and bronchopulmonary (NCI-H727) NET cell models through potentiating cell cycle arrest and preventing DNA damage repair [93].

Similar to improving radiation delivery or the use of radiotherapy-sensitizing agents, a novel avenue of research addressing existing PRRT resistance involves the use of an alternative therapeutic approach by modifying the targeted component of PRRT itself. This has been attempted through the use of a SSTR antagonist rather than an agonist. The use of such an agent is based upon the concept that antagonists are able to bind more effectively to SSTR-expressing tumors, as they are able to bind the SSTR in both the active and inactive state, while agonists can only do so in the active state [94,95,96]. While it is believed that somatostatin antagonists may undergo less tumor internalization with binding, Wild et al. demonstrated that not only could a somatostatin antagonist be utilized for PRRT targeting, but it also had a higher uptake and prolonged duration of radiation delivery than ^177^Lu-DOTATATE therapy as measured by SPECT/CT 3-dimensional voxel-based dosimetry [94]. Reidy-Lagunes et al. were able to determine safe doses of SSTR antagonist PRRT in patients with NETs to avoid the main observed adverse effect, hematologic toxicity [96]. Of the 20 patients, PFS was 21 months with an ORR of 45% with 5% complete response, 40% partial response, and 40% with stable disease [96]. Unfortunately, a clinical trial examining the effect of the PRRT utilizing a somatostatin antagonist, ^177^Lu-OPS201, in NETs was terminated due to a low patient accrual to study. Overall, while PRRT remains an exciting frontier for the targeted treatment of NETs, many studies are already ongoing seeking to improve tumor uptake and response while minimizing toxicities and the addressing observed radioresistance.

### 3.3. Mammalian Target of Rapamycin Inhibitor: Everolimus

Everolimus is a mTOR inhibitor administered as a daily oral pill, approved by the US FDA in 2016 for the treatment of progressive, nonfunctional, well-differentiated NETs of gastrointestinal (GI) or lung origin [28,97]. This approval was based upon the results of the RADIANT-4 trial, demonstrating that treatment with everolimus prolonged median PFS compared to placebo (11.0 months vs. 3.9 months with placebo) in patients in the NET population described above [28]. With regard to overall survival, everolimus reduced risk of death by 36% compared to placebo (HR 0.64, *p* = 0.037), however this benefit was not statistically significant at a second interim analysis of the study [28,98]. Although radiologic responses were infrequently seen (2% in the everolimus arm vs. 1% in the placebo arm), disease stabilization occurred in 81% of patients taking everolimus compared to 64% in the placebo group [28].

Everolimus resistance is believed to largely result from feedback on pathways impacted by its mechanism of action, mTOR inhibition. This can occur at multiple points within the mTOR pathway, including inhibition of mTORC1 shifting the balance towards mTORC2 signaling, thus increasing Akt production, elevation of which is associated with everolimus resistance (Figure 2) [99,100]. Newer-generation mTOR kinase inhibitors including PP242, AZD2014, and OSI-027 exert their effect on both mTORC1 and mTORC2 and could potentially be used to overcome this mechanism of escape [101]. 

Along with the use of broader-spectrum mTOR inhibitors, synergistic targeting of pathways known to be associated with the mTOR pathway has been considered a potential means of addressing everolimus resistance. This includes combining everolimus therapy with the commonly used anti-hyperglycemic agent metformin, known to inhibit Akt signaling through the IGFR-1/IRS-1/PI3K/Akt pathway which can further dampen mTOR signaling [102]. Metformin and everolimus suppressed cell proliferation in panNET cell lines significantly more than everolimus alone (−71% ± 13%, *p* < 0.0001) [102]. The combination of everolimus, metformin, and octreotide is currently being studied in patients with well-differentiated panNETs (NCT02294006) (Table 1). Cixutumumab, an anti-IGF-1R monoclonal antibody, was studied by Dasari et al. in 19 patients with NETs receiving everolimus and octreotide. Seventeen patients had stable disease with RECIST tumor measurements reduced by 1 to 22% [103]. Median PFS was 43.6 months and OS was 25.5 months [103]. However, use of cixutumumab was limited by toxicity as patients experienced adverse effects of fatigue, weight loss, hyperlipidemia, hyperglycemia, and mucositis [103]. Additionally, anti-IGF-1R monoclonal antibodies have been studied as monotherapy and have not demonstrated improved tumor responses in larger studies in NETs [103,104,105]. The dual mTOR and PI3K inhibitor, BEZ235, was studied in patients with everolimus-resistant panNETs. Though there was modest anti-tumor activity with a PFS rate of 51.6% at 16 weeks, the PFS rate was not high enough to continue to stage 2 of the study [106]. Additionally, tolerability of the drug was limited with 16 of the 31 patients experiencing grade 3/4 adverse effects that included hyperglycemia, diarrhea, and vomiting [106].

Feedback from mTOR inhibition can also activate the RAF/MEK/ERK pathway resulting in upregulated ERK2 that can reduce the response to everolimus therapy [107,108]. Therefore, combining an mTOR inhibitor with MEK inhibition presents a possible means of combating this feedback response [109,110]. Although this has not yet been well studied within NETs, MEK inhibitors are already approved for other indications including melanoma, thyroid cancer, neurofibroma, and non-small cell lung cancer [111]. 

Genetic mutations at FKB-12 or the FRB domain can significantly impede the action of mTOR inhibitors. FKB-12 is the site where everolimus binds and the FRB domain is the site where the FKB-12 and everolimus complex bind to evoke their action (Figure 2) [101,112]. Since next generation mTOR kinase inhibitors do not bind the FRB domain or FKB-12, they provide a therapeutic approach for overcoming this challenge [101]. In panNETs, mutations in the growth signaling receptor, FGFR4, has been associated with everolimus treatment resistance [113]. While not yet tested within the context of neuroendocrine tumors, selective and pan-FGFR inhibitors are an active area of research and are approved in other types of cancer with the potential to be a target for NETs alone or in synergistic combination with everolimus in patients with everolimus-resistant disease [114]. 

An additional potential approach to overcoming everolimus resistance in NETs could involve a drug interruption (holiday) allowing for restoring of sensitivity. Vandamme et al. demonstrated that in everolimus-resistant panNET cell lines QGP-1 and BON-1, a drug holiday resulted in restored sensitivity to everolimus after 10-12 weeks [107]. This study further demonstrated that in these same cells, utilization of drugs that target the PI3K-AKT-mTOR pathway, such as OSI-027, AZD2014, and NVP-BEZ235, inhibited cell proliferation in everolimus-resistant cells [107]. Although this has not previously been studied, a possible strategy to overcome everolimus resistance could involve an alternating therapy regimen allowing for re-sensitization of the mTOR pathway or targeting another related pathway such as PI3K. More clinical research is needed in the various approaches for overcoming everolimus-resistance in order to optimize its disease stabilizing effect.

### 3.4. Receptor Tyrosine Kinase Inhibitors: Sunitinib

Sunitinib is a small-molecule, multi-targeted receptor tyrosine kinase (RTK) with activity against multiple RTKs including vascular endothelial growth factor receptors (VEGFR type 1 and 2), platelet-derived growth factor receptors (PDGFR-alpha and PDGFR-beta), stem cell factor receptor (KIT) [115]. Sunitinib was approved in 2011 by the US FDA for the treatment of progressive, well-differentiated pancreatic neuroendocrine tumors based upon the results of a pivotal phase III clinical trial (A6181111, NCT00428597) which was terminated early based upon early indications of PFS benefit favoring sunitinib over placebo [115,116,117]. In a retrospective, blinded analysis of the trial data performed by independent third-party radiologists, PFS was assessed as 12.6 (95% CI 11.1–20.6) months and 5.8 (95% CI 3.8–7.2) months for the sunitinib and placebo arms, respectively (HR 0.32; 95% CI 0.16–0.55; *p* < 0.0001) [118]. The activity of sunitinib in panNETs is thought to be attributable to the highly vascularized nature of these tumors which express a many pro-angiogenic molecules, including VEGF, fibroblast growth factor (FGF), and PDGF [3].

Development of secondary resistance to sunitinib as evidenced by progression after initial disease stabilization or response to this agent is frequently seen in pancreatic neuroendocrine tumors [119]. Multiple molecular mechanisms are thought to account for the development of this resistance including the activation of alternative pro-angiogenic pathways by panNET cells, hypoxia-induced changes to the tumor micro-environment, lysosomal sequestration of sunitinib, and the induction by sunitinib of tumor autophagy, thus promoting tumor cell survival and treatment resistance (Figure 2) [120,121,122,123,124]. 

Activation of alternative pro-angiogenic pathways is thought to be mediated through the hypoxia-induced activation of HIF-1α, resulting in activation of pathways associated with the expression of proangiogenic factors, such as FGFs, ephrins, angiopoietins, c-MET (hepatocyte growth factor receptor) or epithelial-mesenchymal transition (EMT) induction [125,126,127]. Alterations in the local tumor microenvironment leading to sunitinib resistance are multiple and include the hypoxia-induced recruitment of bone marrow-derived cells including vascular progenitors and pro-angiogenic myeloid cells and the recruitment of inflammatory tumor-associated macrophages. The latter may result in the secretion of multiple angiogenic and pro-inflammatory cytokines including IL-6, TNF-α, and IL-1 [128,129,130,131]. Lysosomal sequestration of sunitinib occurs as a result of the hydrophobic and weakly basic nature of the drug, allowing it to cross the liposomal membrane freely where it is protonated and thus sequestered and eventually degraded [132]. Finally, treatment with sunitinib has been demonstrated in panNET cell lines to induce autophagy, a process that may favor the development treatment resistance and more aggressive tumor behavior [120,121,133].

Addressing the activation of alternative pro-angiogenic pathways through the use of alternative TKIs with distinct and in some cases broader targets represents a particularly promising area of research. Surufatinib, is a TKI with activity against VEGFR1, VEGFR2, VEGFR3, FGFR1 and CSF-1R was demonstrated in the SANET-p and SANET-ep trials to have significant PFS benefit in both pancreatic and extrapancreatic GEP NETs [134,135]. Other TKIs with broad activity are also under active study at this time including cabozantinib (targeted against VEGFR1-3, MET, AXL) and lenvatinib (targeted against multiple VEGFR1-3, KIT and RET tyrosine kinases) (Table 1) [136,137]. Results of lenvatinib in grade 1 and 2 GEP-NETS, including those previously treated with sunitinib, have been particularly promising as reported in the TALENT trial in which ORR of 44.2% (panNET) and 16.4% (GI-NET) and median duration of response of 19.9 (8.4–30.8) and 33.9 (10.6–38.3) months in the panNET and GI-NET groups, respectively, was seen [138]. 

Disrupting cellular autophagy and lysosomal function have additionally been researched in the preclinical setting for the purpose of addressing TKI resistance and improving efficacy of these agents. In Wiedmer et al., addition of the autophagy inhibitor chloroquine to sunitinib treatment decreased tumor viability and reduced tumor burden in the Rip1Tag2 transgenic panNET mouse models compared to either treatment alone [120]. Lysosomal sequestration of sunitinib has been successfully targeted in vitro in renal and colon cancer cell lines utilizing bafilomycin A1, a specific inhibitor of vacuolar type-H^+^-ATPase and in breast cancer cell lines utilizing SB02024, a highly potent and selective inhibitor of vacuolar protein sorting 34 (Vps34) [139,140]. Further in vitro studies in panNET models have demonstrated that autophagic cell death can be promoted through epigenetic modification utilizing the combination of the pan-deacetylase inhibitor Panobinostat and bafilomycin [141].

### 3.5. Chemotherapy

Temozolomide is an orally active alkylating agent prodrug. Its cytotoxic mechanism of action involves delivering methyl groups to purine bases of DNA [142]. Temozolomide, and temozolomide-based chemotherapy combinations (particularly in combination with capecitabine), have demonstrated significant activity in panNETs. In a prospective, randomized phase II trial of temozolomide compared to temozolomide and capecitabine in patients with advanced panNETs, a median PFS benefit was seen with the combination at 22.7 months versus 14.4 months for temozolomide alone (HR = 0.58, *p* = 0.023). With regard to ORR, there were similar at 33.3% for the combination versus 27.8% for temozolomide alone (*p* = 0.47) [143].

Development of resistance by panNETs to treatment with temozolomide-based chemotherapy is well documented, however the mechanism of this is poorly understood [144]. In other tumors, particularly glioblastoma multiforme (GBM), expression of O^6^-methylguanine-DNA-methyltransferase (MGMT) by tumor cells allows for repair of temozolomide-mediated DNA damage, thus conferring treatment resistance [145]. As MGMT expression is suppressed by methylation of the MGMT promoter, de-methylation of the MGMT promoter signifies an epigenetically derived resistance mechanism, linked to decreased survival and responsiveness to temozolomide in patients with GBM [146]. MGMT promoter methylation status is of uncertain significance with regard to alkylating agent treatment response in panNETs, with conflicting results demonstrated in several retrospective studies [147,148].

While the role of MGMT in the development of resistance to temozolomide-based chemotherapy remains unclear, multiple clinical trials attempting to augment the effect of this treatment are ongoing. Several of these studies are focused on combining temozolomide with other DNA-damaging agents including Lutetium Lu 177 Dotatate (NCT05247905), the poly ADP ribose polymerase (PARP) inhibitors talazoparib (NCT05142241), and Yttrium-90 radioisotope treatment (NCT04339036) (Table 1). 

### 3.6. Immune Therapies for GEP-NETs

Immune therapies utilizing immune checkpoint inhibitors (PD-1/PD-L1, CTLA-4), cellular and acellular vaccines, and chimeric antigen receptor (CAR) T-cell therapies have recently revolutionized the treatment of multiple hematologic and solid tumor malignancies. Unfortunately, these same promising results have yet to be reproduced for the treatment of GEP-NETs. In a subgroup analysis of the CA209-538 prospective clinical trial evaluating the combination of the CTLA-4/PD-1 inhibitors ipilimumab/nivolumab in patients with advanced rare cancers, treatment with these immune checkpoint inhibitors demonstrated an encouraging ORR of 24% in 29 patients with neuroendocrine neoplasms [149]. Further studies have yet to replicate these promising results and a retrospective analysis conducted by Al-Toubah et al. reported a more modest ORR of 14.7%, with responses entirely limited to those patients with poorly differentiated NECs [150]. While the monotherapy activity of immune checkpoint inhibitors has thus far been disappointing, the potential synergistic combination of TKI agents with immune checkpoint inhibitors for the treatment of NETs is a highly active area of clinical research. Combinations under active study include the addition of cabozantinib to ipilimumab and nivolumab (NCT04079712), lenvatinib to pembrolizumab (NCT03290079), and surufatinib to tislelizumab (NCT04579757) (Table 1).

Bi-specific T-cell engagers (BiTEs) are a novel immune therapy consisting of a synthetic bispecific monoclonal antibody with two single-chain variable fragments, one binding to a T cell via the CD3 receptor, and the other to a tumor cell via a tumor specific molecule. Through the co-localization of cytotoxic T cells to neoplastic cells, these therapies have demonstrated exciting immune activation and anti-tumor activity in both hematologic and solid tumor malignancies [151,152]. Considering the limited efficacy of immune therapies thus far observed in GEP-NETs, BiTEs represent an encouraging avenue of research to overcome the immune-evading nature of these diseases. One such BiTE, XmAb18087 (tidutamab), a humanized anti–SSTR2, anti-CD3 bispecific antibody, demonstrated significant T cell activation and SSTR2^+^ cell depletion within in vitro and in vivo NET models [153]. The results of this preclinical study informed the development of a phase 1 clinical trial evaluating XmAb18087 in advanced grade 1 and 2 NETs (NCT03411915). Preliminary results of this study were presented at the 2020 North American Neuroendocrine Tumor Society Annual Symposium, where it was reported that XmAb18087 treatment resulted in a best response of stable disease in 6/14 (43%) evaluable patients, and treatment was accompanied by an acceptable adverse event profile and sustained T-cell activation in peripheral blood [154].

Similarly to BiTEs, CAR-T cell therapies have shown remarkable efficacy in the treatment of multiple hematologic malignancies [155,156,157,158]. While CAR-T cells have yet to be evaluated in clinical trials for patients with NETs, CDH17-targeted CAR-T cells have shown excellent preclinical in vitro and in vivo activity against gastrointestinal cancer xenografts and autochthonous mouse models including NETs with minimal toxicity to non-malignant tissues [159]. While still early in development, the potential use of CDH17-targeted CAR-T cells represents an exciting potential avenue to enhance immune therapies for GEP-NETs in the future.

## 4. Conclusions

GEP-NETs are a heterogenous group of rare neoplasms with an increasing incidence and few effective therapeutic options for patients with advanced or metastatic disease. This lack of available therapies is only made more challenging in these difficult to treat malignancies by the frequent and sometimes rapid development of resistance to the common therapies utilized. This article summarizes the current research landscape to elucidate the various molecular mechanisms GEP-NETs utilized for the development of treatment resistance. 

Notable mechanisms include epigenetic modification and protein expression alteration, SSTR internalization or phosphorylation, alteration of alternative cellular feedback or DNA repair pathways, mutations to drug binding sites, lysosomal sequestration of drugs, and drug-induced cellular autophagy. While these mechanisms result in significant challenges in treatment GEP-NET patients, they also provide important information for possible means of optimizing the treatment effect or re-inducing sensitivity of existing therapies while also providing a potential guidance in the development of novel therapeutic agents. 

Further research into these various mechanisms of treatment resistance and means of overcoming them could allow for the development of improved and personalized treatment approaches to patient’s unique tumors. Several clinical trials are underway to optimize tumor response to each of the common therapies, offering promising improvements in the future management of GEP-NETs. 

## Figures and Tables

**Figure 1 cancers-14-04769-f001:**
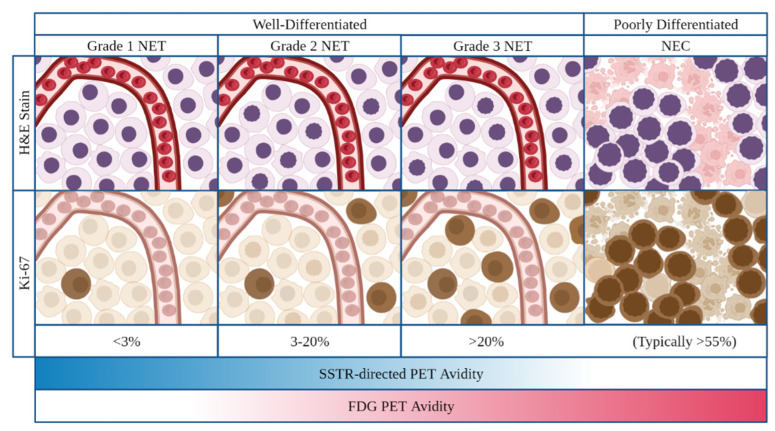
Classification and grading criteria for neuroendocrine neoplasms of the gastrointestinal tract as per 2019 World Health Organization criteria. On H&E and IHC staining, NETs (well-differentiated) demonstrate preserved organoid pattern and infrequent mitotic activity while NECs (poorly differentiated) demonstrate loss of organoid morphology, necrosis, and typically high levels of mitotic activity. On imaging, increasing tumor grade frequently demonstrates an inverse correlation with SSTR-directed PET activity and a direct correlation with FDG-PET activity. Abrreviations: NET neuroendocrine tumor; NEC neuroenocrine carcinoma, H&E hematoxylin and eosin; PET positron emission tomography, FDG fluorodeoxyglucose.

**Figure 2 cancers-14-04769-f002:**
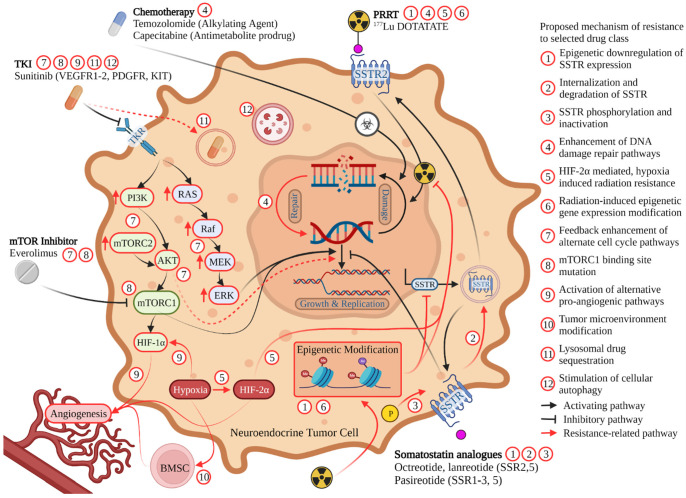
Mechanism of action for commonly utilized therapies for the treatment of GEP-NETs. Described mechanisms of treatment resistance are denoted by red arrows with specific resistance mechanism to each therapeutic class denoted by a corresponding numbered red circle. Abbreviations: SSTR somatostatin receptor, PRRT peptide receptor radionuclide therapy, mTOR mammalian target of rapamycin, TKI tyrosine kinase inhibitor, VEGFR, vascular endothelial growth factor receptor; PDGFR, platelet-derived growth factor receptor; KIT stem cell growth factor receptor, TKR tyrosine kinase receptor, BMSC bone marrow-derived stem cells.

**Table 1 cancers-14-04769-t001:** Active or planned trials to overcome resistance or optimize anti-tumor response in GEP-NETs.

Identifier	Experimental Treatment Arm	Phase	Status
SSTR directed therapies			
NCT05249114	177Lu-DOTATATE + cabozantinib	1b	Not yet recruiting
NCT05053854	177Lu-DOTATATE + Talazoparib	1	Recruiting
NCT04194125	177Lu-DOTATOC, capecitabine, temozolomide	2	Recruiting
NCT05247905	177Lu-DOTATATE, capecitabine, temozolomide	2	Not yet recruiting
NCT03044977	Combined 131I-MIBG + 90Y-DOTATOC	1	Active, NR
NCT05153772	212Pb-DOTAMTATE (PRRT naive)	2	Recruiting
NCT03466216	212Pb-DOTAMTATE (after prior PRRT)	1	Recruiting
NCT05178693	68Ga-DOTATATE + ASTX727	1	Not yet recruiting
NCT05477576	225Ac-DOTATATE (after prior PRRT)	1b/3	Recruiting
mTOR directed therapies			
NCT02294006	Everolimus, metformin, octreotide	2	Active, NR
NCT01229943	Everolimus, octreotide acetate, bevacizumab	2	Active, NR
NCT03950609	Everolimus with Lenvatinib	2	Recruiting
Receptor Tyrosine Kinase and HIF2α directed therapies			
NCT02589821	Surufatinib (panNETs)	3	Active, NR
NCT02588170	Surufatinib (non panNET GEP-NETs)	3	Active, NR
NCT04579757	Surufatinib + Tislelizumab	1/2	Recruiting
NCT05165407	Surufatinib, IBI310, Sintilimab	2	Recruiting
NCT03375320	Cabozantinib	3	Recruiting
NCT05048901	Cabozantinib + Lanreotide	1/2	Not yet recruiting
NCT04893785	Cabozantinib + Temozolomide	2	Recruiting
NCT04197310	Cabozantinib + Nivolumab	2	Recruiting
NCT04079712	Cabozantinib, Nivolumab, Ipilimumuab	2	Active, NR
NCT03290079	Pembrolizumab and Lenvatinib	2	Recruiting
NCT01841736	Pazopanib	2	Active, NR
NCT02399215	Nintedanib	2	Active, NR
NCT04924075	Belzutifan monotherapy (panNETs)	2	Recruiting
Chemotherapy			
NCT03217097	MGMT Status: Response to Alkylating Agents	n/a	Active, NR
NCT05142241	Temozolomide + Talazoparib (rare tumors)	2	Recruiting

Abbreviations: GEP-NET gastroenteropancreatic neuroendocrine tumors, SSTR somatostatin receptor, 177Lu Lutetium-177, DOTATATE tetraxetan octreotate, DOTATOC tetraxetan Tyr3-octreotide, 131I Iodine-131, MIBG meta-iodobenzylguanidine, 90Y Yttrium-90, 212Pb Lead-212, DOTAMTATE DOTAM (chelator) octreotate, PRRT peptide receptor radionuclide therapy, ASTX727 oral decitabine and cedazuridine, 225Ac Actinium 225, mTOR mammalian target of rapamycin, HIF2α hypoxia-inducible factor 2 alpha, panNET Pancreatic neuroendocrine tumor, GEP-NETs gastroenteropancreatic neuroendocrine tumor, MGMT O6-methylguanine-DNA methyltransferase, NR not recruiting.

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
