# Peer review of "Therapy Resistant Gastroenteropancreatic Neuroendocrine Tumors"

_cancers, 2022, doi:10.3390/cancers14194769_

Round 1

Reviewer 1 Report

This is a relatively novel review addressing mechanisms of resistance to currently emplyed therapeutic modalities for gastroentropancreatic tumors. The authoes need though to clearly distinguish the development of resistance (primary or secondary) to the several therapeutic modalities from the effect of concomitant or additional treatments to obtain a therapeutic response by reducing tumor load. In addition, the use of therapeutic approaches that have other mechanisms of action such as telotristat to my opinion is not a mean to overacome resistance to SSA but to obtain a synergistic effects. Similarly using radiosotope emitters with different properties is an alternative therapeutic approach as other molecular targeted therapies and/or chemotherapeutic schemes. The authors should also make an effort to make a more clear distinction of the mechanisms in respect of the anti-secretory and antiproliferative effect particulalry on SSA. There are several prospective (NETTER 1 and CLARINET FORTE) and many retrospective trials that need to be mentioned for the SSA effect. In addition, I feel that there is a lot of information in respect of the RCT that have modified the treatment of these tumors that could be shortened and more focused.

The authors address several pathophysiological/molecular pathways but could also include some other works proposing alternative mechanisms (Pedraza-Arevalo S, 2021, Molecular Oncology)

Specific comments

Line 37: the therapeutic options for these tumors have been expanded substantially the last decade compared to only SSA and chemotherapy that were available before. There are other means of chemotherapy that can also be used and a disrinction between primary and secondary resistance could be made

Line 69: the increase described is mainly locorigional disease whereas metastatic disease remained stable. Stomach NET are much more common that described particulalrly type 1

Line 76: rare primaries that are not related to gastrointestinal system

Figure 1: not highly relevant neither informative

Line 161: Check ESMO guidelines to include all parameters that need to be considered

Line 210: define resistance clinical (carcinoid syndrome - bowel movements fro TELESTAR study) antisecretory, anti-prolifetaive

Line 261: the dose of octreotide used for comparison was not 30 but 40 mg

Line 269: telotristat has synergistic effect and not overcoming resistance as no anti-proliferative effect

Please see previous comment regarding distinguishing resistance to a therapeutic modality per se rather than employing a synergistic drug

The length could be reduced and make more obvious the relative novelty of the study that is the mechanisms of resitance and future developments

Author Response

Reviewer 1, thank you so much for your thoughtful review comments. Please find responses below:

"The authors need though to clearly distinguish the development of resistance (primary or secondary) to the several therapeutic modalities from the effect of concomitant or additional treatments to obtain a therapeutic response by reducing tumor load. In addition, the use of therapeutic approaches that have other mechanisms of action such as telotristat to my opinion is not a mean to overcome resistance to SSA but to obtain a synergistic effects. Similarly using radioisotope emitters with different properties is an alternative therapeutic approach as other molecular targeted therapies and/or chemotherapeutic schemes."

- I agree with the recommendations made in the above comment. I have gone through the manuscript and altered the language to address where addressing observed resistance to an agent can be considered through the synergistic combination with another agent or through the use of an alternative yet related agent.

"The authors should also make an effort to make a more clear distinction of the mechanisms in respect of the anti-secretory and antiproliferative effect particularly on SSA."

-Agree with above comment. Section added to SSA section describing in more detail the ant-secretory and antiproliferative effects of SSAs and the mechanisms of action through SSTR activation

"There are several prospective (NETTER 1 and CLARINET FORTE) and many retrospective trials that need to be mentioned for the SSA effect. In addition, I feel that there is a lot of information in respect of the RCT that have modified the treatment of these tumors that could be shortened and more focused."

-I have tried to trim the above sections to focus as recommended. That said, with the requests/recommendations of the reviewer and editor the revised version is longer than prior.

"The authors address several pathophysiological/molecular pathways but could also include some other works proposing alternative mechanisms (Pedraza-Arevalo S, 2021, Molecular Oncology)"

-Thank you for the recommended inclusion. This was added (Line 247)

Line 37: the therapeutic options for these tumors have been expanded substantially the last decade compared to only SSA and chemotherapy that were available before. There are other means of chemotherapy that can also be used and a distinction between primary and secondary resistance could be made

-Added a description of the expanded therapeutic options and a section directly in the introduction describing the distinction between primary and secondary resistance as recommended.

Line 69: the increase described is mainly locorigional disease whereas metastatic disease remained stable. Stomach NET are much more common that described particulalrly type 1

-Both recommendations acknowledged and added to the body of the text

Line 76: rare primaries that are not related to gastrointestinal system

-Removed mention of these rare primaries unrelated to GEP-NETs

Figure 1: not highly relevant neither informative

-Figure 1 was included as a part of the introduction trying to graphically summarize what can be a complex histologic classification system with the frequently observed imaging findings associated with different stage/differentiation disease. That said, if the reviewer and editor would prefer the removal of this figure, would be happy to do so.

Line 161: Check ESMO guidelines to include all parameters that need to be considered

-Reviewed ESMO guidelines related to observed genetic syndromes in GEPNETs (particularly panNETs), no specific guidelines. Checked NANETs and ENETs and the recommended genetic syndrome considerations included those listed in the text (MEN1, VHL, TS, NF1).

Line 210: define resistance clinical (carcinoid syndrome - bowel movements fro TELESTAR study) anti-secretory, antiprolifetaive

-Resistance defined both conceptually and as defined in the Telestar study

Line 261: the dose of octreotide used for comparison was not 30 but 40 mg

-Corrected

Line 269: telotristat has synergistic effect and not overcoming resistance as no anti-proliferative effect

-Changed wording and language in the text to reflect this fact and that the utility of telotristat is synergistic with SSA to address the secretory effects in carcinoid syndrome

Please see previous comment regarding distinguishing resistance to a therapeutic modality per se rather than employing a synergistic drug

-Agreed, and as above tried to address the language throughout the text to distinguish these

The length could be reduced and make more obvious the relative novelty of the study that is the mechanisms of resistance and future developments

-I attempted to shorten and streamline as described above. That said with the requested changes and additions (please note added immunotherapy section at the end) the length of the review has actually increased with these revisions.

Reviewer 2 Report

I have to congratulate with you for the extensiveness of the research you have done and the amount of data you are providing in this review.

Line 62: May be useful to distinguish GEP-NETs that are resistant to first-line therapy in the first place vs GEP-NETs that develop in a second moment a resistance to the therapies that were effective in a first moment.

Line 86: It could be highlighted that G2 is a very heterogeneous category with low-grade G2s and high-grade G2s that may have a significantly different outcome (ex. Hauck 2016 Scandinavian J Endocr); may be also interesting to highlight this point of view.

Line 141: I would shortly underline that these functioning NETs also have specific blood markers and specific testing that can be viable for diagnosis.

Line 261: I think here you should underline that in this study from Wolin et al. the patients in the studies were all experiencing carcinoid symptoms refractory to 1st gen SSAs.

It may also be useful to address, when there is viable evidence, the presence or absence of predictors of response for every specific therapy (the ones where you can find the most are first generation SSAs and PRRT I think); in this discussion you could also evaluate whether is useful or not to highlight blood testing (like NETtest or others) that may help to provide the most effective therapy to a specific patient.

Author Response

Reviewer 2, thank you so much for your thoughtful comments and review. Please find my attached responses below:

Line 62: May be useful to distinguish GEP-NETs that are resistant to first-line therapy in the first place vs GEP-NETs that develop in a second moment a resistance to the therapies that were effective in a first moment.

-Agreed. This concept was added in the introduction (line 66) and within the body of the paper (multiple locations)

Line 86: It could be highlighted that G2 is a very heterogeneous category with low-grade G2s and high-grade G2s that may have a significantly different outcome (ex. Hauck 2016 Scandinavian J Endocr); may be also interesting to highlight this point of view.

-Agreed, heterogenous nature of Grade 2 disease added (line 92)

Line 141: I would shortly underline that these functioning NETs also have specific blood markers and specific testing that can be viable for diagnosis.

-Agreed and added to the bottom of that paragraph

Line 261: I think here you should underline that in this study from Wolin et al. the patients in the studies were all experiencing carcinoid symptoms refractory to 1st gen SSAs.

-Agreed, description of the study modified to underline that this was in refractory carcinoid 

"It may also be useful to address, when there is viable evidence, the presence or absence of predictors of response for every specific therapy (the ones where you can find the most are first generation SSAs and PRRT I think); in this discussion you could also evaluate whether is useful or not to highlight blood testing (like NETtest or others) that may help to provide the most effective therapy to a specific patient."

-I added sections to the SSA and PRRT sections describing observed predictors to response to treatment. I attempted to add a section describing the potential utility of NETtest and the possibilities of peptide receptor radiotherapy prediction quotient, but the discussion of these becomes quickly long and complex and potentially outside the scope of this review. If requested by the reviewer and editor, an additional section (perhaps as a discussion piece at the end) could be included, but again this seems outside the scope of the review.